# The Impact of Uncertainty on Trade: The Case for a Small Port

Noor Zahirah Mohd Sidek [1,*] , Bhuk Kiranantawat [2] and Martusorn Khaengkhan [2]

1   Department of Economics, Faculty of Business Management, Universiti Teknologi MARA,
    Sungai Petani 08400, Kedah, Malaysia
2   College of Logistics and Supply Chain, Suan Sunandha Rajabhat University, Dusit, Bangkok 10300, Thailand
*   Correspondence: nzahirah@uitm.edu.my

**Abstract:** In the present paper, we show how uncertainty emanating from fluctuations in economic uncertainty, news-based uncertainty, and geopolitical risks affect the number of containers exported from Thailand via Penang Port, Malaysia. Our sample extends from January 2009 to May 2020 from three main entry points in the Northern Peninsular Malaysia–Thailand Border: Padang Besar, Surat Thani, and Bukit Kayu Hitam. Two modes of transportation of containers are mainly used for export purposes, namely, road and rai. This study examines the nonlinear effect of uncertainty on trade by employing a two-regime Markov regime-switching approach. The empirical results show that, overall, uncertainty significantly affects the movement of containers in the high-uncertainty regime. Therefore, small ports must continue to diversify their client base to cushion the impact of fluctuations in global trade due to uncertainty.

**Keywords:** Malaysia–Thailand cross-border trade; economic policy uncertainty; geopolitical news; economic news

## 1. Introduction

In recent years, studies have shown that economic uncertainty tends to dampen and reduce the volume of trade as a result of the delayed investment, increased precautionary savings, and decreased consumption, which later leads to economic slowdown (Dogah 2021; Grier and Smallwood 2013). Although the impact of uncertainty on the stock market, financial market, and firm performance has been extensively examined (Gupta and Wohar 2017; Balcilar et al. 2016; Kollias et al. 2017), the effect of uncertainty on trade has yet to be subjected to serious empirical investigation. Therefore, there are still gray areas regarding how uncertainty affects trade. Previously, the main problem with assessing the impact of uncertainty on trade was the lack of a reliable dataset that allows for comparative analysis for countries. With the availability of new datasets to capture news, geopolitical events, and economic policy uncertainty, an in-depth analysis on trade uncertainty can be conducted to comprehend its impact.

The assessment of the impact of uncertainty on trade is expected to have important implications for at least two (2) reasons. First, uncertainty may trigger a decline in both trade volume and growth. For example, uncertainty arising from trade tension between the United States and China may later lead to financial war or any other existing economic-related cooperation involving both countries. If the trade war persists, other countries would be able to adjust given the knowledge (news) and extent of the impact of the trade war on other countries. For example, the trade war results in disturbances in oil prices where oil prices plummeted during the trade war since China is one of the world's biggest oil consumers and the United States has large oil reserves, and the recent discovery of shale oil provides abundant oil supply. This information would be vital for the identification of risk-inducing shocks and how to assuage such shocks. Second, examining the different types of uncertainty serves as an effective learning mechanism for institutional investors, traders, and risk managers in their risk assessments. Discerning how different types of

uncertainty would allow them to strategize and mitigate such risks on their trading position by diversifying trade with other countries.

Malaysia and Thailand's trade with the US and China accounts for more than 10 percent of their total trade, which mainly consists of semi-finished products; raw materials, such as wood; unrefined petroleum; and finished goods. From January to June 2020, Malaysia's trade value with China and the US was approximately USD 23 billion. Moreover, Malaysia's export increases to China and the US's is 46.8% and 27.6%, respectively, compared to 2019. This indicates that, in a volatile situation, such as COVID-19, it is difficult to predict the impact of trade uncertainty (Malaysia External Trade Development Corporation 2020). Thus, it is crucial that the impact of trade uncertainty be studied to better predict the outcome and mitigate affected trade.

According to the IMF, global growth is expected to contract by −4.9 percent in 2020 (WEO 2020). The deadly COVID-19 pandemic coupled with the US–China trade war, which has persisted since 2017 until the outbreak of COVID-19 in December 2019, has brought the global economy to a possible V-shaped recession. The United States projected growth is −8.0 percent, which is much lower than China's projected growth of 1.0 percent in 2020 (WEO 2020). The increase in tariffs arising from the US–China trade war has not only reduced trade between the two countries, but is expected to disrupt global supply chains, consumers having to pay higher prices due to higher tariffs, and dent businesses in connection with China and the United States. The COVID-19 outbreak worsened the situation, but somewhat muted the trade war. In July 2020, the United States regulators began to push for more transparency from Chinese companies with share trading in the United States. For the past decade, several Chinese-owned companies used reverse mergers with dormant United States companies, which were later made public on the United States' stock exchanges. Disputes arising from the trade war may persist in another form, which is a financial war between these two economic powerhouses.

The objective of this study is to examine the impact of economic uncertainty on container throughputs in Penang Port and how it affects trade in Malaysia and Thailand. The Penang Port is used in this study due to several interesting reasons. It serves as a feeder port of raw-material exports from Southern Thailand and the Northern part of Malaysia to other parts of Asia, mainly to China, Hong Kong, and Japan. Another vital addition is the period of study that covers the US–China trade war period; the extended operating hours at the Malaysia–Thailand border from 18 h to 24 h daily at the Immigration, Customs, Quarantine, and Security (ICQS) Complex at Bukit Kayu Hitam, Malaysia; and Custom, Immigration and Quarantine (CIQ), Sadao, Thailand from 18 June 2019 to 17 June 2020, and covers the onset of the COVID-19 pandemic. In addition, Penang's total exports and imports of goods in 2020 amounts to RM 500.4 billion, equivalent to 31.8% of total Malaysia's trade with a trade surplus of RM 124.4 billion (Lee 2021). Penang also contributed 68.2% of the total RM 189.3 billion exports of electronic integrated circuits and 44% of RM 88.6 billion exports in electric and electronic products in Malaysia. Hence, despite being a small port, Penang Port is responsible for exports to China, the US, Singapore, Taiwan, Hong Kong, Japan, South Korea, Germany, and Vietnam.

This study discusses the impact of economic uncertainty from the perspective of news-based economic policy uncertainty, and uncertainty due to geopolitical risks in a nonlinear framework. A nonlinear relationship implies a non-direct relationship between variables where changes in one variable may not be proportional to changes in another variable. We argue that the nonlinear method is more appropriate to capture these uncertainties. The subsequent section overviews how the United States–China trade war impacted the trade in South-East Asian countries, focusing on Thailand and Malaysia. Section 3 reviews the selected literature pertaining to uncertainty arising from events and disturbances and the impact on the economy. Section 4 reviews the estimation methodology. The penultimate section discusses the results and policy recommendations, and the final section provides an overall conclusion and way forward for the supply chain ecosystem.

## 2. News (US–China Trade War) and Its Impact on South-East Asian Trade

The US–China Trade War was initiated when the United States initiated an initial investigation in late 2017 using the Trade Act of 1974 and Trade Expansion Act of 1962 for alleged unfair trade practices. The trade tension began on 22 January 2018 when the United States imposed global safeguard measures to imports of solar panels and washing machines from China. In addition, the United States invoked Section 232 of the Trade Expansion Act 1962 to levy additional tariffs on aluminum and steel imports on national security grounds. The situation was further aggravated when the European Union (EU) imposed retaliatory measures in March 2018 on 23 iron and steel products, which was later followed by Turkey in May 2018.

Being a small, open economy with a heavy reliance on exports, both Malaysia and Thailand were subjected to the pros and cons of the ongoing trade war. In the case of Malaysia, for example, the United States–China trade war created a trade diversion where outsourcing activities that were normally undertaken by China for the United States moved to Malaysia. For example, Poh Huat Resources Holding Berhad received a major outsourcing contract from the United States in place of China. The Malaysian Furniture Council estimated a 20 percent increase in furniture exports to the United States in the next three years (The Edge 2019). Thailand, on the other hand, faced a downturn in terms of exports for rubberwood to China due to the reduction in the production of furniture for United States imports for China. Another trade diversion was palm oil. Prior to the trade war, China's cooking oil consumption was mainly fulfilled by soy oil imports from the United States. The trade war diverted cooking imports for palm oil from Malaysia and Indonesia. Between July 2017 and Jun 2018, palm oil exports from Malaysia to China increased by 6.7 percent. Investments in and the relocation of both multinational companies from the United States and China to Malaysia, Vietnam, and Thailand may occur if production costs and increases in tariff due to the trade war persist. More diverted investments are expected if the trade war persists. For example, Apple Inc. is expected to move 15–30 percent of production from China to a cheaper location in South-East Asia (Nikkei Asian Review, 19 Jun 2019). In Malaysia, for example, the first quarter of 2019 saw a surge in foreign investments of 74.2 percent in the manufacturing sector or approximately USD 7.75 billion, where USD 2.74 and USD 1.12 billion are from the United States and China, respectively.

In July 2018, China was the second-largest trade partner for Malaysia after Singapore, whilst the United States came third. Malaysia's main exports to the United States and China account for approximately 22.8 percent of the total exports during Jun 2017–Jun 2018. By July 2018–Jun 2019, Malaysia's exports to the United States and China accounted for 38.7 and 53.5 percent, respectively, which mainly constituted electric and electronic products. Despite the 2017–2019 US–China trade wars, exports from Malaysia and Thailand still record growth (see Table 1). Malaysia's exports to the US and China recorded growth of around 5.9 percent or approximately USD 32.33 billion. Robust exports were contributed to by LNG, metal, steel, seafood, electric and electronics, machinery, and optical equipment. In general, trade with the United States remains in the cases of export and import from Malaysia and Thailand to the United States. Table 1 shows the export and import growth for pre-trade conflict (July 2017–June 2018) and during trade conflict (July 2018–June 2019). The export growth increased from 8.6 to 10.5 percent from Thailand. In the case of Malaysia, exports to the United States declined from 11.1 to 2.1 percent for the same period. In the case of imports, Thailand's import growth from the United States increased from 2.5 to 19.0 percent, whilst Malaysia saw a smaller increase in import growth from 6.8 to 7.4 percent.

**Table 1.** Trade growth of selected markets to the USA.

| Markets | Export Growth | | Import Growth | |
|---|---|---|---|---|
| | Pre-Trade Conflict | Ongoing Trade Conflict * | Pre-Trade Conflict | Ongoing Trade Conflict |
| | July 2017–June 2018 % Growth | July 2018–June 2019 % Growth | July 2017–June 2018 % Growth | July 2018–June 2019 % Growth |
| Vietnam * | 5.9 | 21.1 | −13.8 | 17.3 |
| Singapore | 21.3 | 19.4 | 13.7 | 17.3 |
| Taiwan * | 7.7 | 14.7 | 1.0 | 20.3 |
| Thailand | 8.6 | 10.5 | 2.5 | 19.0 |
| Malaysia | 11.1 | 2.1 | 6.8 | 7.4 |
| China | 11.6 | 1.0 | 12.4 | −18.7 |

Source: GTA IHS/DOSM; Notes: * sorted by export growth during the ongoing trade conflict period.

Table 2 presents the export and import growth from selected markets to China. Thailand's export growth to China plummeted from 13.7 percent pre-trade conflict to −6.6 percent during trade conflicts. Import growth saw a lower reduction from 11.4 to 3.8 percent. In the case of Malaysia, export growth contracted from 19.0 percent pre-crisis to 4.4 percent during the trade war. On a similar note, import growth dropped from 15.1 to 1.7 percent for Malaysia.

**Table 2.** Trade growth of selected markets to China.

| Markets | Export Growth | | Import Growth | |
|---|---|---|---|---|
| | Pre-Trade Conflict | Ongoing Trade Conflict | Pre-Trade Conflict | Ongoing Trade Conflict |
| | July 2017–June 2018 % Growth | July 2018–June 2019 % Growth | July 2017–June 2018 % Growth | July 2018–June 2019 % Growth |
| Vietnam * | 53.7 | 23.2 | 23.9 | 13.1 |
| Malaysia | 19.0 | 4.4 | 15.1 | 1.7 |
| Taiwan * | 17.9 | −0.2 | 13.1 | 8.9 |
| Thailand | 13.7 | −6.6 | 11.4 | 3.8 |
| Singapore | 7.6 | −7.0 | 13.1 | 0.8 |
| USA | 9.7 | −20.0 | 9.4 | −3.2 |

Source: GTA IHS/DOSM, IPSOM 2020; Notes: * sorted by export growth during the ongoing trade conflict period.

Figure 1 illustrates the entry of containers for export from South Thailand to Penang Port. Most of the exports from Thailand via Penang Port would be shipped to China, Japan, Taiwan, Hong Kong, and the Philippines. Exports from Thailand are mainly rubberwood and other rubber products. The preferred mode of entry is via rail through the Padang Besar entry point, which is the only means of rail access from Thailand to Malaysia. The mode of entry using the road is less popular due to relatively higher costs compared to rail. The use of Padang Besar rail as an entry point increased from 53.44 percent in 2016 to 59.96 percent in 2019. The road-entry point via Bukit Kayu Hitam gradually increased from the period of 2009 to 2017, but recorded a slight decrease in 2018–2019. In 2016, the total container entry via Bukit Kayu Hitam entry point was 35.82 percent, but by 2019 this was reduced to only 27.66 percent. Penang Port recorded a sharp decline in exports of rubberwood and other rubber products from Thailand to China. The number of containers (TEUs) from South Thailand to China via Penang Port substantially reduced during the trade war. This reduction was due to a contraction in the demand by the United States for wood molding products from China. During the trade war, the United States imposed an anti-dumping tax of 183.6 percent and an anti-subsidy tax of up to 194.9 percent on imported wood products from China. The Thailand government imposed the Agreed Export Tonnage Scheme (AETS) in 2018, which led to a reduction in export and rubber coupled with a decline in rubber prices to USD 60–70; the closure of furniture production in China due to renovation to comply with China's new environmental standards substantially impacted

the industries in South Thailand and Penang Port. With the COVID-19 pandemic and global recession, the number of trade activities is expected to contract further in 2020.

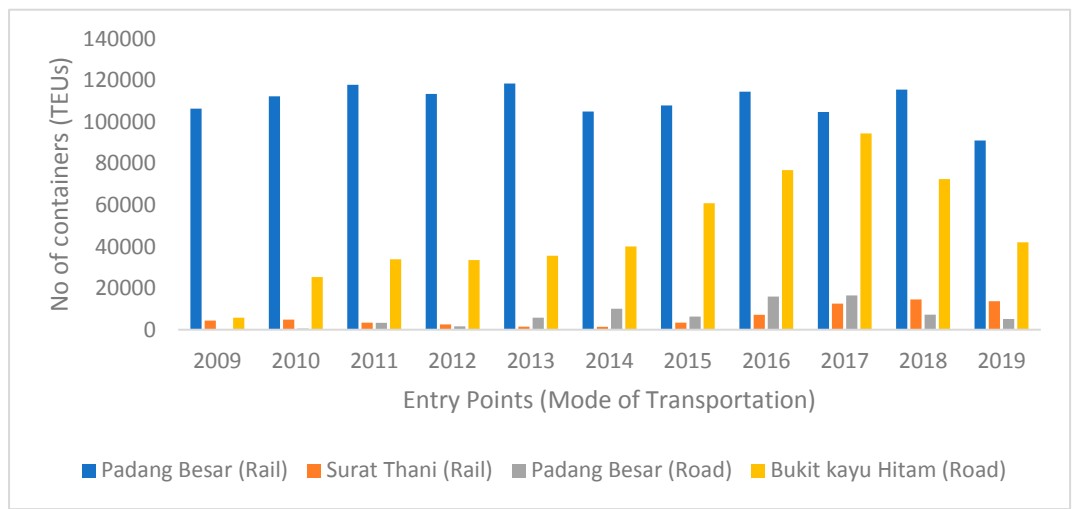

**Figure 1.** Mode of transportation for containers from Thailand to Penang Port for export. Source: PPSB (2020).

## 3. Review of the Selected Literature

International trade has brought about unprecedented growth and development to a majority of countries involved in exports and imports. Over the years, the metamorphosis of international trade has taken the form of globalization, reduction in tariff and non-tariff barriers, and more bilateral and multilateral free-trade agreements, all of which aim to promote greater export and import rates. The rise of China as an export powerhouse prompted several countries to engage in FTAs with China or established their own trading blocs, such as the European Union (EU), North American Free Trade Area (NAFTA), ASEAN Free Trade Area (AFTA), amongst others. For example, FTA between China and the European Free Trade Association (EFTA) shows a high export volume from Iceland and Switzerland to China post FTA (Kristjánsdóttir et al. 2022). Uncertainty arising from domestic or international events causing changes or disturbances in the economy, economic policies, polity, political conflict, trade, or stock market would substantially affect economic and corporate behavior, especially in terms of asset allocation and investment decisions (Bouoiyour et al. 2019; Caldara and Iacoviello 2022; Wolfers and Zitzewitz 2009). Exchange-rate risks also give rise to uncertainty and have been subjected to rigorous empirical testing with varying conclusions (for example, Broll et al. 2020; Lin et al. 2020; Klaassen 2004; Perée and Steinherr 1989). Uncertainty tends to increase in the event of elections, which either leads to a change in the ruling government or otherwise. Ruling political party transitions would result in changes in policies since winning an election means an expectation from the public to observe noticeable changes in economic or social policies that would affect businesses (inter alia Lee and Lee 2018; Li et al. 2021) and trade (Hill et al. 2019). The main channel for uncertainty is via the news. Uncertainty arising from overall stock-price changes is often captured by the volatility index (VIX) and volatility index futures (NVIXs), uncertainty from economic policies is normally captured via the economic policy uncertainty index (EPU), and geopolitical risks are mainly captured by the geopolitical risks index (GPR).

Policy uncertainty can be defined as '*the economic risk associated with undefined future government policies and regulatory frameworks*' (Al-Thaqeb and Algharabali 2019). This can postpone corporate spending and investments and consumers' spending. Baker et al. (2016) mentioned that uncertainty in government policies spiked just after the 2008 financial crisis, which was caused by uncertainty from corporate and consumers involving changing government fiscal, economic, and healthcare. Hence, it can be implied that when there is a

considerable crisis leading to recession, policy uncertainty will impede recovery due to a delay or decrease in consumer and corporate investment and expenditure. As uncertainty has a considerable effect on the government, business, and individual spending and investment, this prompted many researchers to create concrete measures to determine uncertainty, especially in economic and policy uncertainty. As a result, the volatility index (VIX) was created to measure volatility in the stock market. It has been used by businesses and firms to measure uncertainty in the market. The major drawback is that it is only applicable to the established market as it measures market depth and liquidity (Al-Thaqeb and Algharabali 2019). A more recently developed measure is the news-based volatility index (NIVX). It was created by utilizing texts from the Wall Street Journal to measure uncertainty (Manela and Moreira 2017). However, it does not provide the overall picture of uncertainty as it only captures uncertainty based on news.

The economic policy uncertainty index (EPU) was created to capture uncertainty encompassing policy, market, news, and economy (Baker et al. 2016). Thus, it is a more accurate index used to measure overall uncertainty in policy, which can help determine uncertainty and forecast the effect of uncertainty. From the study conducted by Huang and Luk (2020), EPU was used in China, in which media was heavily censored to measure economic uncertainty. The result revealed that despite heavy media censorship, EPU accurately determined the effect of high uncertainty, which coincided with a decrease in economic activities. Gupta and Sun (2020) also proved that EPU can be used to accurately predict economic policy uncertainty in BRIC and other developing countries. Furthermore, EPU is proven to be an ideal index that serves as a basis for further research on new-based uncertainty. As shown in the research of Ardia et al. (2019), new-based sentiment indices were developed, built on EPU to capture new-based sentiment values. These values were used to predict short-term US industrial production with great accuracy. Suitably, EPU was employed in this research to study the effect of uncertainty on trade, which can be used to forecast the impact of uncertainty on trade and export in Malaysia and Thailand.

Trade policy uncertainty is another aspect that this research addressed. Trade policy has become a crucial factor that drives international trade due to rapid globalization. The ongoing US–China Trade War and the imminent change in trade policies due to the 2020 US election increased trade policy uncertainty, which could have a considerable impact on international trade. Caldara et al. (2020) found that unexpected changes in trade policies leading to high trade policy uncertainty can decrease business investment and economic activities in the US. In addition, it was also found that an increase in trade policy uncertainty can slow down Chinese companies' investment in new foreign markets (Crowley et al. 2018). Therefore, in this study, it was important to investigate how trade policy uncertainty affected trade and export in Malaysia and Thailand.

Geopolitical risk, as defined by Caldara and Iacoviello (2022), is the 'risk associated with wars, terrorist acts, and tensions between states that affect the normal and peaceful course of international relations. Geopolitical risks can have an adverse effect on the economy as they contribute to a decline in economic activities, reduce stock returns, and cause outflows of capital (Caldara and Iacoviello 2022). Moreover, geopolitical risks can lead to considerable uncertainty causing an large and sudden impact on oil price, which affects trade (Brandt and Gao 2019). Gupta et al. (2019) showed that geopolitical risks in general negatively affect trade flows for 164 countries. Similar findings are echoed in Yang et al. (2022); Cheng and Chiu (2018); and Bouoiyour et al. (2019). In addition, geopolitical risks arising from trade wars would likely lead to higher tariffs and later reduced trade between the countries involved. In the worse cases, the tension might permeate to third-party countries involved indirectly with trade to both countries at war. Hence, it is important to measure the geopolitical risks as they can inherently affect trade and investment. The geopolitical risks index (GPR) is a widely accepted index largely used to capture geopolitical risks systematically. In this study, GPR was utilized to measure geopolitical risks and the impact on trade and export in Malaysia and Thailand.

### 3.1. Empirical Estimation Method and Data

Table 3 offers the preliminary statistics for all variables used in the study. Table 4 presents no strong correlation between the variables, which reduces the possibility of multicollinearity in the estimation.

**Table 3.** Descriptive statistics.

|              | Mean      | Maximum     | Minimum  | Std. Dev. | Observations |
|--------------|-----------|-------------|----------|-----------|--------------|
| BKH_ROAD     | 4112.7740 | 10,556.0000 | 149.0000 | 2264.7630 | 137          |
| CHINA_EPU    | 229.0956  | 852.0525    | 59.4412  | 157.2609  | 133          |
| CHINA_NEWS   | 279.9389  | 970.8299    | 26.1441  | 230.9545  | 133          |
| CHINA_TPU    | 155.4398  | 413.8014    | 83.5258  | 60.5949   | 138          |
| GEPU_CURRENT | 159.2704  | 425.6608    | 81.8783  | 65.5911   | 138          |
| GPR          | 105.2298  | 370.4247    | 40.5062  | 56.1812   | 138          |
| GPR_CHI      | 111.9746  | 251.2252    | 61.9459  | 32.3835   | 138          |
| GPR_MAL      | 89.8874   | 271.0700    | 22.6284  | 35.6237   | 138          |
| GPR_THA      | 94.4052   | 279.7898    | 35.7548  | 45.0354   | 138          |
| GPR_THR      | 113.2368  | 408.9641    | 31.3702  | 64.7981   | 138          |
| JAP_TPU      | 197.8658  | 699.9738    | 36.5727  | 143.9806  | 127          |
| PB_RAIL      | 9282.0220 | 11,946.0000 | 649.0000 | 1338.5660 | 137          |
| PB_ROAD      | 545.8467  | 1996.0000   | 0        | 531.4399  | 137          |
| ST_RAIL      | 583.8832  | 2069.0000   | 0        | 516.7699  | 137          |
| TP_EMV       | 0.0495    | 0.3540      | 0        | 0.0786    | 127          |
| US_EPU       | 141.8978  | 284.1359    | 63.8773  | 44.7165   | 133          |
| US_TPU       | 137.7891  | 1374.2800   | 10.5643  | 210.1710  | 127          |

### 3.2. Testing for Nonlinearity

Subsequently, we performed a nonlinearity test based on Broock et al. (1996) or the BDS test as a preliminary test to understand the data-generating process of the underlying variables. This pre-testing serves as a basis for the nonlinear method used to understand the impact of uncertainty on trade. The BDS is estimated in the following manner:

$$V_{m,\varepsilon} = \sqrt{T}\, \frac{C_{m,\varepsilon} - C_{1,\varepsilon}^m}{S_{m,\varepsilon}} \tag{1}$$

where $S_{m,\varepsilon}$ is the standard deviation of $\sqrt{T}C_{m,\varepsilon} - C_{1,\varepsilon}^m$. The BDS statistics converge towards $N(0,1)$, so the null hypothesis of *i.i.d.* is rejected when $|V_{m,\varepsilon}| >$ critical values. The BDS test is arguably more powerful compared to other nonlinear techniques, such as the threshold model and Markov-switching model (Ashley and Patterson 2006), to indicate nonlinear behavior in the data. If nonlinearity is established, the next step is to choose an appropriate estimation method to unpack the behavior of the focal variables. Table 5 presents the results of the BDS test that indicate the presence of nonlinearity in the data; hence, this corroborates the use of the Markov regime-switching model for the analysis.

**Table 4.** Correlation.

| | BKH_R | CHINA_E | CHINA_NWS | CHINA_TPU | GEPU_CURRENT | GPR | GPR_CHI | GPR_MAL | GPR_THA | GPR_THR | PB_RAIL | PB_ROAD | ST_RAIL | TP_EMV | US_EPU | US_TPU |
|---|---|---|---|---|---|---|---|---|---|---|---|---|---|---|---|---|
| BKH_R | 1 | | | | | | | | | | | | | | | |
| CHINA_EPU | 0.398 | 1 | | | | | | | | | | | | | | |
| CHINA_NEWS | 0.411 | 0.963 | 1 | | | | | | | | | | | | | |
| CHINA_TPU | 0.386 | 0.883 | 0.898 | 1 | | | | | | | | | | | | |
| GEPU_CURRENT | 0.383 | 0.925 | 0.929 | 0.992 | 1 | | | | | | | | | | | |
| GPR | 0.575 | 0.568 | 0.537 | 0.411 | 0.443 | 1 | | | | | | | | | | |
| CHINA GPR | 0.386 | 0.710 | 0.676 | 0.579 | 0.614 | 0.731 | 1 | | | | | | | | | |
| MALAYSIA GPR | −0.003 | −0.236 | −0.196 | −0.252 | −0.246 | 0.131 | −0.114 | 1 | | | | | | | | |
| THAILAND GPR | −0.238 | −0.389 | −0.346 | −0.426 | −0.432 | −0.221 | −0.169 | 0.234 | 1 | | | | | | | |
| PADANG BESAR RAIL | 0.568 | 0.590 | 0.555 | 0.434 | 0.467 | 0.996 | 0.755 | 0.095 | −0.233 | 1 | | | | | | |
| PADANG BESAR ROAD | 0.037 | 0.191 | 0.184 | 0.205 | 0.210 | 0.014 | 0.098 | −0.087 | −0.220 | 0.037 | 1 | | | | | |
| SURAT THANI_RAIL | 0.726 | 0.270 | 0.303 | 0.249 | 0.251 | 0.357 | 0.236 | 0.041 | 0.027 | 0.345 | −0.054 | 1 | | | | |
| TP_EMV | 0.536 | 0.805 | 0.766 | 0.650 | 0.684 | 0.597 | 0.672 | −0.303 | −0.337 | 0.618 | 0.201 | 0.346 | 1 | | | |
| US_EPU | 0.368 | 0.764 | 0.708 | 0.618 | 0.647 | 0.660 | 0.815 | −0.275 | −0.307 | 0.691 | 0.150 | 0.148 | 0.736 | 1 | | |
| US_TPU | 0.059 | 0.403 | 0.421 | 0.710 | 0.645 | 0.040 | 0.231 | −0.295 | −0.333 | 0.063 | 0.123 | −0.080 | 0.248 | 0.291 | 1 | |
| CHINA GPR | 0.316 | 0.831 | 0.770 | 0.735 | 0.759 | 0.630 | 0.751 | −0.211 | −0.323 | 0.663 | 0.149 | 0.126 | 0.699 | 0.878 | 0.416 | 1 |

**Table 5.** BDS test for nonlinearity.

| | BDS Statistics | | | | |
|---|---|---|---|---|---|
| | *m* = 2 | *m* = 3 | *m* = 4 | *m* = 5 | *m* = 6 |
| BUKIT KAYU HITAM—RAIL | 0.1274 *** | 0.2168 *** | 0.2857 *** | 0.3302 *** | 0.3562 *** |
| CHINA_EPU | 0.1503 *** | 0.2521 *** | 0.3177 *** | 0.3577 *** | 0.3829 *** |
| CHINA_NEWS | 0.1422 *** | 0.2406 *** | 0.3012 *** | 0.3381 *** | 0.3597 *** |
| CHINA_TPU | 0.1153 *** | 0.1912 *** | 0.2374 *** | 0.2665 *** | 0.2900 *** |
| GEPU_CURRENT | 0.1265 *** | 0.2099 *** | 0.2615 *** | 0.2953 *** | 0.3210 *** |
| GPR | 0.0807 *** | 0.1302 *** | 0.1646 *** | 0.1816 *** | 0.1859 *** |
| CHINA GPR | 0.1073 *** | 0.1771 *** | 0.2255 *** | 0.2599 *** | 0.2781 *** |
| MALAYSIA GPR | 0.0290 *** | 0.0536 *** | 0.0652 *** | 0.0683 *** | 0.0687 *** |
| THAILAND GPR | 0.0568 *** | 0.0904 *** | 0.1046 *** | 0.1109 *** | 0.1065 *** |
| GPR BROAD | 0.0853 | 0.1422 | 0.1718 | 0.1875 | 0.1941 |
| PADANG BESAR RAIL | 0.0004 | −0.0279 *** | −0.0275 ** | −0.0270 ** | −0.0265 ** |
| PADANG BESAR ROAD | 0.0965 *** | 0.1621 *** | 0.1971 *** | 0.2120 *** | 0.2216 *** |
| BUKIT KAYU HITAM ROAD | 0.1274 *** | 0.2168 *** | 0.2857 *** | 0.3302 *** | 0.3562 *** |
| SURAT THANI_RAIL | 0.1435 *** | 0.2523 *** | 0.3238 *** | 0.3719 *** | 0.4056 *** |
| TP_EMV | 0.1380 *** | 0.2314 *** | 0.2973 *** | 0.3400 *** | 0.3658 *** |
| US_EPU | 0.0397 *** | 0.0662 *** | 0.0759 *** | 0.0770 *** | 0.0703 *** |
| US_TPU | 0.1436 *** | 0.2417 *** | 0.3014 *** | 0.3382 *** | 0.3680 *** |

Notes: *** and ** represent significance at 1% and 5% levels.

### 3.3. Markov Regime-Switching Model

Given that nonlinearity is suggested based on the results of the BDS test in Section 4.1, this section proceeds to describe the Markov regime-switching model to further understand the reaction of trade in the face of uncertainty. The Markov regime-switching model is normally used to capture nonlinearity emanating from different behaviors, different reactions towards certain variables or events, or due to structural breaks. As a result, the data-generating process is no longer linear and may be broken down into one or more subsamples, regimes, or states. The advantage of using the Markov regime-switching model is that the timing of the switch is not predetermined, so that the timing of such regime switches is unknown but determined via estimation.

The Markov regime-switching model based on Hamilton (1989) proposed an estimation method for parameters of different regimes, subsamples, or states by allowing an intercept term, the slope of the coefficient, and variance to be state-dependent. The timing of the shift from one regime to another does not affect the parameter estimation. In the spirit of the work conducted on the finances of the stock market and oil price behavior (see, for example, Li et al. 2021; Chen and Shen 2007; Reboredo 2010), we assumed the existence of at least two possible regimes: (i) a regime where uncertainty was low and (ii) a regime with higher uncertainty. Low uncertainty is associated with low volatility and higher uncertainty is linked to high volatility. Therefore, the probability for the transition between regimes can be expressed as follows:

$$y_t = \begin{cases} X_t\beta_1 + u_t \ (u_t|s_t) \sim NID\left(0, \sum_1 \ldots s_t = 1\right) \\ X_t\beta_1 + u_t \ (u_t|s_t) \sim NID\left(0, \sum_M \ldots s_t = M\right) \end{cases} \tag{2}$$

Hamilton (1989) describes the auto-regressive model as follows:

$$Y_t = \mu_{st} + \sum_{j=1}^{p} \varnothing_{jst-j}\left(Y_{t-j} - \mu_{st-j}\right) + \sigma_{st}\varepsilon_t \tag{3}$$

A rearrangement of the equation can be written as:

$$\Delta Y_t - \mu(s_t) = \varnothing_1\left(\Delta Y_{t-1} - \mu(s_{t-1})\right) + \cdots\cdots\cdots + \varnothing_p(\Delta Y_{t-p} - \mu\left(s_{t-p}\right) + u_t \tag{3a}$$

where the state variable is assumed to follow the first-order Markov chain with a transition probability

$$P = \begin{pmatrix} P_r(s_t = 1 | s_{t-1} = 1) = p_{11}, P_r(s_t = 1 | s_{t-1} = 2) = p_{21} \\ P_r(s_t = 2 | s_{t-1} = 1) = p_{12}, P_r(s_t = 2 | s_{t-1} = 2) = p_{22} \end{pmatrix} = \begin{pmatrix} 1 - p_{12}, \; p_{21} \\ p_{12}, \; 1 - p_{21} \end{pmatrix} \quad (4)$$

The estimation of the transition probabilities $p_{ij}$ is based on the maximum likelihood.

The conditional probability density function for $Y_t$ given state variables $s_t$, $s_{t-1}$ and the previous observations $F_{t-1} = \{y_{t-1}, y_{t-2}, \ldots\}$ is presented as follows:

$$f(y_t \mid s_t, s_{t-1}, F_{t-1}) = \frac{1}{\sqrt{2\pi\sigma_{s_t}^2}} \exp\left\{ -\frac{[y_t - \mu_{s_t} - \varphi(y_{t-1} - \mu_{s_{t-1}})]^2}{2\sigma_{s_t}^2} \right\} \quad (5)$$

which is also the likelihood value for an observed value or $s_t$ in a given regime since $\mu_t = y_t - \mu_{s_t} - \varphi(y_{t-1} - \mu_{s_{t-1}})) \sim NID(0, \sigma_{s_t}^2)$. The joint probability density function for variables $y_t$, $s_t$, $s_{t-1}$, given the past information of $F_{t-1}$, is $f(y_t \mid s_t, s_{t-1}, F_{t-1}) = f(y_t \mid s_t, s_{t-1}, F_{t-1}) P(s_t, s_{t-1}, F_{t-1})$. The log-likelihood function is maximized with respect to some unknown parameters, such as:

$$l_t(\theta) = \sum_{t=1}^{T} l_t(\theta) \quad (6)$$

where

$$l_t(\theta) = \log\left[ \sum_{s_t=0}^{1} \sum_{s_{t-1}=0}^{1} f(y_t \mid s_t, s_{t-1}, F_{t-1}) P(s_t, s_{t-1}, F_{t-1}) \right] \quad (7)$$

and $\theta = (p, q, \varphi, \mu_0, \mu_1, \sigma_0^2, \sigma_1^2)$, and the transition probabilities are $p : P(S_t = 0 \mid S_{t-1} = 0)$ and $q : P(S_t = 1 \mid S_{t-1} = 1)$.

Another interesting feature of the Markov regime-switching model is that it allows the estimation of duration in addition to the transition from one regime to another. The transition probabilities are used to calculate the expected length of time in one regime, $j$. Assuming $D_j$ is the number of periods or length of time in regime $j$, the probability to stay in the $k$ period in regime $j$ can be written as follows:

$$P(D_j = k) = p_{jj}^{k-1}(1 - p_{jj}) \quad (8)$$

where the expected duration of that particular regime $j$ written is as follows:

$$E(D_j) = \sum_{k=0}^{\infty} kP(D_j = k) = \frac{1}{1 - p_{jj}} \quad (9)$$

where $p_{11} = p$ and $p_{22} = q$.

Based on the above discussion, the empirical model is loosely based on Reboredo (2010) to incorporate the impact of uncertainty on trade.

$$TEU_t = \alpha_0 + \beta_1 U + \mu_t \quad (10)$$

where $TEU_t$ is the number of containers calculated based on the standard twenty-foot units arriving at the port, $U$ captures uncertainty, and $\mu$ is the residual of the regression.

## 4. Data and Sources

This study used data from Penang Port Sdn Bhd (PPSB) to capture the movement of exports from Southern Thailand to various Asian countries. The uniqueness of PPSB is due to its capacity to cater to both domestic and neighboring countries' containers. In addition, the distance of PPSB compared to Bangkok Modern Terminal or Laem Chabang

ports is much closer, hence reducing logistic costs. The data on the numbers of containers in standard twenty-foot equivalent units (TEUs) were used to represent units of the cargo capacity of a standard container used for shipping goods. The data were obtained from PPSB. The length of data was dictated by the availability of data from PPSB. The data were collected from three (3) entry points from the Northern Border of Malaysia where export from Southern Peninsular Thailand to Asian countries is transport via Padang Besar (rail and road), Surat Thani (rail), and Bukit Kayu Hitam (road).

The economic policy uncertainty was developed by Baker et al. (2016). To date, this database offers a variety of uncertainty measures ranging from economic policy uncertainty for global and country levels, trade policy uncertainty indices, US equity market volatility index, financial stress indicator, firm-level political risk, world uncertainty index, geopolitical risk, immigration-related risk, and firm-level uncertainty. More recently, they offer a few more indices, which include twitter-based uncertainty, infectious diseases EMV, and COVID-19-induced economic uncertainty. The GDP-weighted average national of EPU (GEPU) index is used as a proxy to capture global economic policy uncertainty. This index is a weighted average of 21 countries' economic policy uncertainty indices, which include Australia, Brazil, Canada, Chile, China, Colombia, France, Germany, India, Ireland, Italy, Japan, Mexico, the Netherlands, Russia, South Korea, Spain, Sweden, the United Kingdom, and the United States. Another alternative proxy for US trade uncertainty is the trade policy share of the US equity market volatility (Trade Policy EMV Fraction). This index measures the percentage of articles on equity market volatility in mainstream American newspapers that discuss trade matters. The geopolitical risk is based on Caldara and Iacoviello (2022) who constructed a monthly index to capture the words related to any form of political tensions from 11 international newspapers.

### 4.1. Empirical Results and Discussion

This section is divided into three sub-sections. The first sub-section offer the benchmark results followed by two other sections using other proxies to capture uncertainty to ensure the robustness of the results.

Benchmark Results

The Markov regime-switching used in this study allowed for switching in the intercept and relaxed the assumption of constant variances across two different regimes. Table 6 presents the estimation results obtained for Markov regime-switching models for four different types of transportation across three entry points, which include (i) Padang Besar—rail, (ii) Surat Thani—rail, (iii) Padang Besar—road, and (iv) Bukit Kayu Hitam—road. The Markov regime-switching model splits the sample into low and high uncertainty periods where such uncertainty arises from changes in economic policies, news, trade policies, and geopolitical risks.

The results presented in Table 7 show that, in general, China's economic policy uncertainty (*China EPU*) affects the number of container movements for export purposes from Thailand to Malaysia's Penang Port. For rail transport, the movement of containers via both Padang Besar and Surat Thani entry points are affected by China's economic policy uncertainty during both low and high uncertainty cases. No significant evidence is found in the case of low uncertainty for Padang Besar's entry point via rail. In the case of road transportation, China's economic policy uncertainty is significant during the high-uncertainty regime for Padang Besar, and in the case of Bukit Kayu Hitam, it is significant during the low-uncertainty period. Economic policy uncertainty in the US (*US EPU*), however, does not have a statistically significant effect on the number of container movements between Thailand–Malaysia via these entry points. These results lend support to the fact that most of Thailand's export via Penang Port, Malaysia is for Asia's market, which includes China, Hong Kong, Taiwan, and Japan. China's trade policy uncertainty (TPU) is significant in both low- and high-uncertainty regimes for rail transport.

**Table 6.** Regime-switching dynamic regression for number of containers (TEUs) arriving at Penang Port via different modes of transportation.

| | Padang Besar—Rail | | | | Surat Thani—Rail | | | |
|---|---|---|---|---|---|---|---|---|
| | **Model 1** | **Model 2** | **Model 3** | **Model 4** | **Model 1** | **Model 2** | **Model 3** | **Model 4** |
| **Regime 1—Low Uncertainty** | | | | | | | | |
| $U$ | 1.6919 | 2.8161 | 4.6307 ** | 0.4909 | 1.4440 *** | 1.8938 | 2.4926 *** | 1.0208 *** |
| | (3.0339) | (2.2905) | (1.9356) | (1.0764) | (0.1983) | (1.0082) | (0.4345) | (0.2828) |
| $c$ | 8060.641 *** | 9424.197 *** | 8784.645 *** | 8503.953 *** | 662.7572 *** | 983.7904 *** | −24.8764 | 230.6212 *** |
| | (1053.279) | (402.7895) | (317.6218) | (426.1973) | (88.2953) | (162.2668) | (63.527) | (28.6850) |
| **Regime 2—High Uncertainty** | | | | | | | | |
| $U$ | 1.4885 ** | −4.9955 | −9.6512 ** | 0.8853 ** | 0.7628 ** | 0.7456 | 3.9532 *** | 0.4763 *** |
| | (0.5921) | (4.6572) | (4.3145) | (0.5417) | (0.3530) | (0.6056) | (0.8200) | (0.1334) |
| $c$ | 9395.731 *** | 9316.373 *** | 8392.554 *** | 9648.486 *** | 156.3656 *** | 192.9191 ** | 520.6865 *** | 1074.410 *** |
| | (241.1602) | (597.3881) | (1316.605) | (200.1047) | (56.4154) | (83.5067) | (168.2427) | (65.6824) |
| $\sigma_1$ | 6.7666 *** | 6.7967 *** | 6.9398 *** | 6.7978 *** | 5.2775 *** | 5.4888 *** | 5.4058 *** | 5.3638 *** |
| | (0.0951) | (0.0749) | (0.0692) | (0.0830) | (0.0647) | (0.0642) | (0.0632) | (0.0653) |
| **Transition Probabilities** | | | | | | | | |
| $p11$ | 1.4691 | 2.7301 *** | 3.7986 *** | 1.9245 ** | 3.3131 *** | 4.4144 *** | 4.6855 *** | 4.9167 *** |
| | (1.5129) | (0.8499) | (0.7574) | (1.1585) | (0.8561) | (1.4490) | (1.0856) | (1.1963) |
| $P21$ | −2.5124 *** | −2.3297 *** | -1.7261 | −2.5539 *** | −3.9883 *** | −4.9413 *** | −3.2978 *** | −4.2754 *** |
| | (0.9200) | (0.7849) | (1.4972) | (0.8677) | (0.7708) | (1.2140) | (0.9168) | (1.4914) |
| LL | −1105.926 | −1107.308 | −1156.810 | −1058.606 | −903.6399 | −924.4807 | −944.6841 | −867.3438 |
| AIC | 16.7357 | 16.7565 | 16.9899 | 16.7811 | 13.6938 | 14.007 | 13.8932 | 13.7691 |
| SC | 16.8878 | 16.9086 | 17.1391 | 16.9379 | 13.8459 | 14.1593 | 14.0424 | 13.9259 |
| HQC | 16.7975 | 16.8183 | 17.0505 | 16.8448 | 13.7556 | 14.0690 | 13.9538 | 13.8328 |
| | Padang Besar—Road | | | | Bukit Kayu Hitam—Road | | | |
| | **Model 1** | **Model 2** | **Model 3** | **Model 4** | **Model 1** | **Model 2** | **Model 3** | **Model 4** |
| **Regime 1—Low Uncertainty** | | | | | | | | |
| $U$ | 1.3848 | 0.0291 | 0.9969 ** | 0.4846 *** | 4.4860 *** | −6.9153 | 5.1380 | 3.0098 *** |
| | (0.9852) | (0.7323) | (0.5042) | (0.1453) | (0.7825) | (4.0047) | (5.0471) | (0.6875) |
| $c$ | 1026.369 ** | 320.8186 *** | 164.4022 | 247.2118 *** | 1860.872 *** | 7093.293 *** | 5869.038 *** | 2393.372 *** |
| | (219.4071) | (112.3924) | (86.0782) | (40.8256) | (205.7368) | (631.9894) | (815.3219) | (161.6400) |
| **Regime 2—High Uncertainty** | | | | | | | | |
| $U$ | 0.7534 *** | −2.3039 | −0.9770 | −0.7551 | 3.6169 | 0.0891 | 15.0113 *** | 0.9273 |
| | (0.1749) | (1.4663) | (1.3664 | (0.4723) | (2.0992) | (3.7514) | (1.9860) | (1.1155) |
| $c$ | 136.1150 ** | 1656.345 *** | 1492.807 *** | 1435.324 *** | 5798.258 *** | 2433.166 *** | 676.7385 ** | 6532.997 *** |
| | (53.5060) | (213.4814) | (229.7346) | (92.4823) | (551.6591) | (536.8640) | (330.8139) | (280.0799) |
| $\sigma_1$ | 5.6443 *** | 5.7620 *** | 5.7343 *** | 5.7216 *** | 7.0762 *** | 7.1886 *** | 7.1106 *** | 7.1287 *** |
| | (0.0784) | (0.0718) | (0.0678) | (0.0694) | (0.0638) | (0.0637) | (0.0627) | (0.0656) |
| **Transition Probabilities** | | | | | | | | |
| $p11$ | 1.6802 *** | 3.5817 *** | 3.5503 *** | 3.3873 *** | 4.7421 *** | 4.7922 *** | 3.4957 *** | 4.6724 *** |
| | (0.5497) | (0.6857) | (0.6433) | (0.6366) | (1.0758) | (1.295) | (0.8920) | (1.0798) |
| $P21$ | −3.0292 *** | −2.1521 *** | −2.1017 *** | −2.0636 *** | −3.4811 *** | −4.9145 *** | −4.7247 *** | −3.4692 *** |
| | (0.5564) | (0.6525) | (0.6209) | (0.6081) | (0.895) | (1.2426) | (1.0759) | (0.8928) |
| LL | −967.1459 | −973.1707 | −999.7684 | −926.6911 | −1138.720 | −1150.461 | −1177.385 | −1093.834 |
| AIC | 14.6488 | 14.7394 | −14.6973 | 14.7038 | 17.2288 | 17.4054 | 17.2902 | 17.3359 |
| SC | 14.8009 | 14.8915 | 14.8465 | 14.8605 | 17.3810 | 17.5575 | 17.4394 | 17.4927 |
| HQC | 14.7106 | 14.8012 | 14.7579 | 14.7674 | 17.2906 | 17.4672 | 17.3509 | 17.3996 |

Notes: *** and ** denote 1% and 5% significance levels, respectively. The switching variables: Model 1—China EPU; Model 2—US EPU; Model 3—China TPU; Model 4—US TPU.

**Table 7.** Regime-switching dynamic regression for number of containers arriving at Penang Port via different modes of transportation.

| | Padang Besar—Rail | | | Surat Thani—Rail | | |
|---|---|---|---|---|---|---|
| | Model 1 | Model 2 | Model 3 | Model 1 | Model 2 | Model 3 |
| **Regime 1—Low Uncertainty** | | | | | | |
| $U$ | 3.5773 | −9.7101 ** | 1541.396 | 1.0808 *** | 3.5642 *** | 3400.241 *** |
| | (2.2570) | (4.0940) | (2034.298) | (0.1372) | (0.7320) | (274.5706) |
| $c$ | 6796.501 *** | 8441.454 *** | 8457.844 *** | 717.7123 *** | 551.3818 *** | 225.1071 *** |
| | (992.7506) | (1262.841) | (352.2116) | (81.1444) | (160.6190) | (22.0985) |
| **Regime 2—High Uncertainty** | | | | | | |
| $U$ | 1.0254 ** | 4.1762 ** | 3362.042 ** | 0.6035 *** | 2.3803 *** | 3384.961 *** |
| | (0.4053) | (1.7400) | (1528.91) | (0.15174) | (0.4475) | (563.7616) |
| $c$ | 9306.299 *** | 8839.431 *** | 9600.749 *** | 173.9174 *** | −13.4535 *** | 928.4792 *** |
| | (150.2391) | (296.0143) | (178.9183) | (32.6840) | (64.1527) | (80.1067) |
| $\sigma_1$ | 6.7593 *** | 6.9369 *** | 6.7877 *** | 5.2276 *** | 5.3951 *** | 5.2113 *** |
| | (0.0855) | (1.4362) | (0.0807) | (0.0672) | (0.0663) | (0.0824) |
| **Transition Probabilities** | | | | | | |
| $p11$ | 0.4290 | 1.7108 | 1.9466 ** | 2.7312 *** | 3.2008 *** | 2.9472 *** |
| | (0.9024) | (1.4362) | (0.9727) | (0.6979) | (0.9390) | (0.5221) |
| $P21$ | −2.7368 *** | −3.7796 *** | −2.6454 *** | −3.5636 *** | −4.5421 *** | −1.3652 |
| | (0.7695) | (0.7569) | (0.8316) | (0.6410) | (1.1376) | (0.6276) |
| Log−likelihood | −1106.040 | −1156.511 | −1057.506 | −902.8396 | −943.5132 | −870.2932 |
| AIC | 16.7374 | 16.9855 | 16.7638 | 13.6818 | 13.8761 | 13.8156 |
| SC | 16.8895 | 17.1347 | 16.9206 | 13.8339 | 14.0253 | 13.9724 |
| HQC | 16.7992 | 17.0461 | 16.8275 | 13.7436 | 13.9367 | 13.8793 |
| **Regime 1—Low Uncertainty** | | | | | | |
| $U$ | 0.2591 | −0.7644 | −478.5957 | 3.0844 *** | 5.2747 | 10056.19 *** |
| | (0.3975) | (1.3178) | (1907.178) | (0.5007) | (4.8018) | (2065.812) |
| $c$ | 1247.904 *** | 1459.173 *** | 1351.451 *** | 2035.29 *** | 5841.575 *** | 2355.932 *** |
| | (136.0817) | (223.8122) | (96.6075) | (184.6058) | (786.2650) | (157.7376) |
| **Regime 2—High Uncertainty** | | | | | | |
| $U$ | 0.4789 *** | 1.0393 ** | 1599.957 *** | 0.4844 | 14.0891 *** | 4153.947 |
| | (0.1269) | (0.4597) | (359.5000) | (1.3508) | (1.8095) | (2593.197) |
| $c$ | 181.8851 *** | 152.5176 | 226.8441 *** | 6531.349 *** | 761.1131 ** | 6436.281 *** |
| | (49.9344) | (81.3232) | (38.8021) | (467.0315) | (312.0361) | (256.3048) |
| $\sigma_1$ | 5.6925 *** | 5.7290 *** | 5.6946 *** | 7.0823 *** | 7.1003 *** | 7.1065 *** |
| | (0.0728) | (0.0677) | (0.0695) | (0.0638) | (0.0627) | (0.0656) |
| **Transition Probabilities** | | | | | | |
| $p11$ | 2.0114 *** | 2.0963 *** | 2.0712 *** | 4.7229 *** | 3.4910 *** | 4.6732 *** |
| | (0.6298) | (0.6182) | (0.6040) | (1.0762) | (0.8926) | (1.0792) |
| $P21$ | −3.3646 *** | −3.5371 *** | −3.3960 *** | −3.4854 *** | −4.7245 *** | −3.4517 *** |
| | (0.6604) | (0.6404) | (0.6273) | (0.8943) | (1.0759) | (0.8951) |
| Log−likelihood | −967.5098 | −999.2767 | −924.2396 | −1139.609 | −1176.053 | −1091.035 |
| AIC | 14.6542 | 14.6901 | 14.6651 | 17.2422 | 17.2708 | 17.2918 |
| SC | 14.8064 | 14.8393 | 14.8219 | 17.3943 | 17.4200 | 17.4486 |
| HQC | 14.7161 | 14.7508 | 14.7288 | 173040 | 17.3314 | 17.3555 |

Notes: *** and ** denote 1% and 5% significance levels, respectively. The switching variables: Model 1—China News-Based EPU; Model 2—GEPU; Model 3—Trade Policy EMV Fraction.

For rail transport via Padang Besar, high uncertainty negatively affected the number of containers transported through this mode of transportation. In the case of road transport, China's trade policy uncertainty was significant under a low-uncertainty regime in the case of entry via Padang Besar and was significant for a high-uncertainty regime in the case of entry point via Bukit Kayu Hitam. Such a difference may be attributed to the type of goods being transported via each mode of transport and via different entry points. For example, railways are used to transport bulk exports from southern Thailand, which consist of wood and wood products that are used in furniture industries in South China. More fragile and urgent export goods, on the other hand, would use road transport. The United States trade policy uncertainty (US TPU) is significant in high-uncertainty regimes in the case of rail transport via Padang Besar and Surat Thani entry points. US trade policy uncertainty is significant in a low-uncertainty regime for Surat Thani (rail) and road transport via both Padang Besar and Bukit Kayu Hitam. Testing for more than one threshold yielded insignificant results, suggesting that only two regimes were present during the sample period.

### 4.2. Robustness Tests—Alternative Measures of Uncertainty

For robustness, we tested the effect of uncertainty using three alternative measures. The three measures included China's news-based economic policy uncertainty (China News-Based EPU), GDP-weighted average of national EPU indices (*GEPU*) to represent global economic policy uncertainty, and trade policy share of the US equity market volatility (*Trade Policy EMV Fraction*). The findings were fairly consistent with the benchmark results presented in Section 4.1. China's economic policy uncertainty was significant in both regimes for Surat Thani (rail) and was significant in the high-uncertainty regime for Padang Besar (rail and road). For Bukit Kayu Hitam's entry point, China's economic policy uncertainty was significant in the low-volatility regime. As to how global economic policy uncertainty affected the number of container movements, rail transport was significantly affected under both low- and high-uncertainty regimes. In the case of Padang Besar (rail), the impact was negative in the low-uncertainty regime. As for road transport, both Padang Besar's and Bukit Kayu Hitam's entry points experienced a significant impact in the high-uncertainty regime. United States trade policy uncertainty represented by trade policy EMV fraction showed that Surat Thani (rail) was significantly affected in both regimes, Padang Besar (rail and road) entry points were significantly affected in the high-uncertainty regime, and Bukit Kayu Hitam's entry point was significantly affected in the low-uncertainty regime. A plausible explanation for such diverse results may be due to the type of goods exported from Thailand to other countries for the production of final goods, which would then be exported to the United States and other countries in the world.

### 4.3. Robustness Tests—Alternative Measures of Policy Uncertainty in the Form of Geopolitical Risks

Another source of uncertainty is geopolitical risks that could directly or indirectly affect trade. Political uncertainty arising from the US–China trade war, for example, could affect the volume of trade depending on which product and country are affected. Table 8 presents the effect of geopolitical risks on container movements. Geopolitical risks in Thailand (GPR Thailand) only affect the movement of containers from Thailand via Thailand's entry point, Surat Thani (rail). Higher geopolitical risks are associated with a lower number of container movements from Thailand to Malaysia for export purposes through Surat Thani (rail), Padang Besar (road), and Bukit Kayu Hitam (road).

A plausible explanation is when the risk increases, the costs of transportation also increase; hence, a safer, viable, and more competitive mode of transportation would be through Padang Besar (rail). Geopolitical risks in Malaysia affect container movement when risks are relatively low in a negative manner for all entry points except Padang Besar (rail). Such results may be due to the recent instability of the government following the transition from Barisan Nasional (BN) to Pakatan Harapan (PH) and the current Perikatan Nasional (PN). Political instability coupled with volatile policy changes could have dampened trade. However, political instability in Malaysia does not entail aggressive street protests or any other form of violent protest, but is more inclined towards social media war. China's geopolitical risks are also significant in the low-volatility regime where lower risks promote more exports to China for all entry points. As for Bukit Kayu Hitam, the movement of containers also increased during the high-volatility regime, which indicates an increase or switching demand for products from Thailand during the US–China trade war. The global geopolitical risks (GPRs) are significant during high-political-risk situations in all cases except Padang Besar (rail). For Padang Besar (rail), GPR is significant when the risks are low.

The US–China trade war has resulted in changes in the way production and exports occur. For example, since exports of furniture have been limited during the US–China trade war, companies in China may decide to relocate to countries with the source of raw materials. For example, Chinese furniture companies may open new plants in Thailand and export furniture to the US as '*made in Thailand*' instead of '*made in China*' to assuage any export bans from the US.

**Table 8.** Regime-switching dynamic regression for number of containers arriving at Penang Port via different modes of transportation.

| | Padang Besar—Rail | | | | Surat Thani—Rail | | | |
|---|---|---|---|---|---|---|---|---|
| | **Model 1** | **Model 2** | **Model 3** | **Model 4** | **Model 1** | **Model 2** | **Model 3** | **Model 4** |
| **Regime 1—Low Uncertainty** | | | | | | | | |
| $U$ | 14.8811 | 107.4374 *** | 11.4715 | −0.0673 | 0.4847 | −3.9667 ** | 4.3748 *** | 3.6683 *** |
| | (15.3455) | (21.7428) | (27.6069) | (2.4521) | (0.5604) | (1.6268) | (1.1088) | (0.9429) |
| $c$ | 3253.858 | −1664.528 | 6318.247 | 9453.659 *** | 319.8347 *** | 1514.241 *** | 589.3852 *** | 766.4135 *** |
| | (1906.433) | (1735.280) | (3459.406) | (261.8097) | (64.8436) | (123.2746) | (164.4627) | (123.3398) |
| **Regime 2—High Uncertainty** | | | | | | | | |
| $U$ | −4.4634 | −4.2927 | 4.7010 | 304.6323 *** | −3.2913 *** | −0.1907 | −2.6127 | −1.2005 |
| | (2.3880) | (2.4724) | (2.8374) | (79.5825) | (1.2256) | (0.7061) | (1.6422) | (1.1225) |
| $c$ | 9852.230 *** | 9887.763 *** | 8920.490 *** | −17741.13 *** | 1461.771 *** | 288.4942 *** | 516.0702 *** | 365.8604 *** |
| | (237.4110) | (243.1765) | (329.0029) | (6308.632) | (95.229) | (74.8273) | (159.5266) | (96.1773) |
| $\sigma_1$ | 6.9331 *** | 6.8941 *** | 6.9319 *** | 6.8615 *** | 5.4967 *** | 5.5045 *** | 5.4637 *** | 5.4688 *** |
| | (0.0720) | (0.0634) | (0.0649) | (0.0653) | (0.0626) | (74.8273) | (0.0623) | (0.0623) |
| **Transition Probabilities** | | | | | | | | |
| $p11$ | 0.6351 | 1.8138 | 0.5256 | 3.5896 *** | 4.9771 *** | 4.6157 *** | 4.6758 *** | 4.6621 *** |
| | (1.9444) | (1.0360) | (1.3485) | (0.6282) | (1.2180) | (1.3777) | (1.3526) | (1.3581) |
| $P21$ | −3.8820 *** | −3.6955 *** | −3.7873 *** | −0.3852 | −4.6029 *** | −4.9908 *** | −5.0289 *** | −5.0205 *** |
| | (0.7240) | (0.6209) | (0.6220) | (1.6568) | (1.3842) | (1.2122) | (1.200) *** | (1.2028) |
| LL | −1157.316 | −1152.032 | −1158.895 | −1150.082 | −953.0087 | −953.8968 | −848.4135 | −949.1548 |
| AIC | 16.9973 | 16.9201 | 17.0203 | 16.8917 | 14.0147 | 14.0276 | 13.9476 | 13.9584 |
| SC | 17.1465 | 17.0693 | 17.1695 | 17.0409 | 14.1639 | 14.1768 | 14.0968 | 14.1076 |
| HQC | 17.0579 | 16.9808 | 17.0809 | 16.9523 | 14.0753 | 14.0883 | 14.0082 | 14.0190 |
| **Regime 1—Low Uncertainty** | | | | | | | | |
| $U$ | 0.1191 | −9.4019 ** | 3.5221 *** | 1.3719 | −6.9956 | −1.0767 *** | 18.830 *** | 11.4329 |
| | (1.2239) | (3.9918) | (0.8601) | (2.7651) | (7.0164) | (4.9493) | (7.0112) | (7.5136) |
| $c$ | 1323.012 *** | 2274.775 *** | −82.4301 | 1208.596 *** | 7327.821 *** | 6143.562 *** | 4589.201 *** | 5509.893 *** |
| | (151.7525) | (410.3890) | (102.5792) | (269.0571) | (695.3123) | (456.8329) | (821.4908) | (830.4401) |
| **Regime 2—High Uncertainty** | | | | | | | | |
| $U$ | −1.8106 ** | −0.1968 | 1.6797 | 3.3478 *** | −9.5735 *** | 3.8857 | 21.8218 *** | 16.7469 *** |
| | (0.7640) | (0.8448) | (2.5606) | (0.7711) | (2.8751) | (4.8107) | (3.9429) | (3.6984) |
| $c$ | 480.2407 *** | 338.2866 *** | 1152.135 *** | −6.3977 | 3822.469 *** | 2078.717 *** | 487.7106 | 1353.502 *** |
| | (75.8017) | (78.6498) | (284.1437) | (80.8842) | (322.7585) | (469.4478) | (450.7068) | (366.2725) |
| $\sigma_1$ | 7.7307 *** | 5.7371 *** | 5.6812 *** | 5.6799 *** | 7.1899 *** | 7.2115 *** | 7.1055 *** | 7.1533 *** |
| | (0.0687) | (0.0662) | (0.0669) | (0.0686) | (0.0632) | (0.0627) | (0.0635) | (0.0635) |
| **Transition Probabilities** | | | | | | | | |
| $p11$ | 2.1497 *** | 2.1192 *** | 3.4748 *** | 2.1087 *** | 3.3964 *** | 4.8563 *** | 3.3611 *** | 3.3567 *** |
| | (0.6583) | (0.6065) | (0.6255) *** | (0.6249) | (0.8405) | (1.2821) | (0.8431) | (0.8445) |
| $P21$ | −3.5895 *** | −3.5510 *** | −2.0528 *** | −3.5488 *** | −3.9954 *** | −4.9311 *** | −3.9700 *** | −3.9625 *** |
| | (0.6583) *** | (0.6289) | (0.6066) | (0.6492) | (0.7680) | (1.2502) | (0.7704) | (0.7729) |
| LL | −999.3483 | −999.8158 | −993.9712 | −992.9206 | −1190.386 | −1187.752 | −1179.965 | −1185.416 |
| AIC | 14.6912 | 14.6980 | 14.6127 | 14.5973 | −17.4800 | 17.4416 | 17.3279 | 17.4075 |
| SC | 14.8404 | 14.8472 | 14.7619 | 14.7465 | −17.6292 | 17.5908 | 17.4771 | 17.5567 |
| HQC | 14.7518 | 14.7586 | 14.6733 | 14.6580 | 17.5407 | 17.5022 | 17.3885 | 17.4681 |

Notes: *** and ** denote 1% and 5% significance levels, respectively. The switching variables: Model 1—GPR Thailand; Model 2—GPR Malaysia; Model 3—GPR China; Model 4—GPR.

## 5. Conclusions

Uncertainty may originate from trade tensions, economic policies due to changes in government, or geopolitical problems or geo-economic fragmentation of the global economy. These uncertainties are expected to have several negative effects on trade.

Uncertainty can lead to a decline in both trade volume and growth. The trade war, for example, could lead to an increase in oil and commodity prices, especially when the countries involved are major exporters or importers. If the trade war leads to economic sanctions, the oil and commodity prices will continue to soar, which in turn escalates uncertainty. The US–China trade war brought about several changes in existing trade constructs, such as trade diversion in the form of outsourcing contracts. Production in

China was diverted to Malaysia and Thailand, which benefits all three parties. Another trade diversion is the increase in exports for palm oil from Malaysia to China as the US imposed trade sanctions on China's importing of vegetable oil from the US. Another form of diversion is investment diversion where the US is expected to move investments from China if the trade war persists.

This paper examines the effect of uncertainty on trade with a specific focus on cross-border trade between Malaysia and Thailand via Penang Port, which represent a case study for a small port. Uncertainty is represented by economic uncertainty, news-based uncertainty, and geopolitical risks, which affect the number of containers for export movements from Thailand to Malaysia. We relied on the two-regime Markov regime-switching model (MSM) technique to capture the impact of uncertainty on the movement of containers during high- and low-uncertainty periods. The movement of containers in our sample covers three entry points that include Surat Thani (rail), Padang Besar (rail and road), and Bukit Kayu Hitam (road). Monthly data spanning from January 2009 to May 2020 were used for the purpose of this study. The results infer that, in general, uncertainty affects the movement of containers, especially during periods of high uncertainty, such as during the US–China trade war or when countries began to lockdown due to COVID-19. Several policy implications can be deduced based on the findings. First, since the movement of goods and services between Malaysia and Thailand is expected to continue in the near future, the authorities in Malaysia should expedite the construction of a transportation hub in Bukit Kayu Hitam. This transportation hub will act as a catalyst to further expand trade between Malaysia and Thailand via land transportation. Once trade volume increases, then it makes sense to have a 24 h operation at the ICQS, Bukit Kayu Hitam. Second, risks arising from geopolitical uncertainty or economic uncertainty that are exogenous in nature are difficult to control. However, countries can mitigate the effects by expanding the number of trading partners, create new exports destinations, and probably engage in more FTAs or trading blocs to cushion the impact of disturbances arising from such uncertainties. The 2022 Russian–Ukraine conflict caused a dramatic increase in commodity and oil prices, resulting in increased inflation and potential recession in less-developed countries. Elevated debt due to the COVID-19 pandemic compounded the effect of rising inflation, leading to greater uncertainty. As countries continue to brace these impacts, small ports would be affected somewhat directly or indirectly. An important strategy that must be adopted by small ports is to continue to diversify their client base and type of exports to help cushion the impact of fluctuations in global trade. Future work could focus on a few ports in Southeast Asia to better comprehend the nature and flow of trade to China and other major trading partners in Southeast Asia.

**Author Contributions:** N.Z.M.S.—Original draft, data curation, methodology, run regression, Review and Editing; B.K.—Literature Review, data collection, introduction, conclusion; M.K.—Literature Review, data collection, conclusion. All authors have read and agreed to the published version of the manuscript.

**Funding:** This work was supported by the SSRU-UiTM Matching Grant 2019–2020.

**Institutional Review Board Statement:** The views expressed in this paper are entirely by the authors and do not necessarily reflect those of UiTM and SSRU.

**Informed Consent Statement:** Not applicable.

**Data Availability Statement:** All data and underlying source codes are available upon request.

**Conflicts of Interest:** The authors declare no conflict of interest.

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
