# Peer review of "The Impact of Uncertainty on Trade: The Case for a Small Port"

_economies, doi:10.3390/economies10080193_

Round 1

Reviewer 1 Report

I accept the article for publication. The article is interesting and well written, explaining the subject well, providing a good setting for the analysis performed. The application of theory is interesting and relevant. Methodology is clearly presented and backs up the subject. The authors should insert one sentence referring to the following article:  Kristjánsdóttir, H., Guðjónsson, S., Óskarsson, G. K. (2022). Free trade agreement (FTA) with China and interaction between exports and imports. Baltic Journal of Economic Studies, 8(1), 1-8. https://doi.org/10.30525/2256-0742/2022-8-1-1-8

Author Response

Dear Reviewer,

Thank you for your insightful comments. Kindly refer to the attachment for details of the response.

Thank you.

Reviewer 2 Report

I don't think this research is significant enough to be published. It uses data that captures the movement of exports from Southern Thailand to various Asian countries but there is no novelty in terms of the methodology or findings. Furthermore, there are no policy implications or lessons presented so it appears to be just a mechanical exercise. There is no recommendation offered to the readers so it is hard to see the point of publishing this research in an academic journal.

Author Response

Dear Reviewer,

Thank you for time to review this manuscript.

Reviewer 3 Report

The paper provide evidence on uncertainty and geopolitical risks affecting number of containers exported from Thailand via pending Port, Malaysia for the period 2009 to 2020 using a monthly data using the Markov regime switching dynamic regression methodology. 

While the study has some prospects and can attract readership. The following are my comments. 

(a)    The motivation for the research needs to be strengthened.

Lots of categorical statement are made with referencing.  

For example, “the deadly COVID-19 pandemic coupled with and to some extent, the US-China trade war which persists from 2017 until the outbreak of COVID-19 in December 2019 has brought the global economy to a possible V-shaped recession”. 

(b)   The study has no objective and needs to be focused.   

At the onset, the show the uncertainty emanating from fluctuations in economic uncertainty, news-based uncertainty, and geopolitical risks affect the number of containers exported from Thailand via Penang Port, Malaysia. 

Then, the author claimed that “Our period of study covers the US-China trade war period, the extended operating hours at the Malaysia-Thailand border from 18 hours to 24 hours daily at the Immigration, Customs, Quarantine, and Security (ICQS) Complex at Bukit Kayu Hitam, Malaysia and Custom, Immigration and Quarantine (CIQ), Sadao, Thailand from 18 June 2019 to 17 June 2020 and covers the onset of the COVID-19 pandemic”.

(c)    The literature review section needs more effort. The authors need to source more literatures.  

(d)   Table 4 correlation matrix is not clear. 

(e)    The conclusion needs to contain policy implications and suggestion for further study.

Author Response

Dear Reviewer,

Thank you for your insightful comments. Kindly refer to the attached file for details of the response.

Thank you.

Reviewer 4 Report

Dear authors, this is a fine contribution to international trade under

uncertainty. Only some minore comments. Please explain the 'non-linear'

effect at the beginning. There is Table 1; behind Table 1 (trade growth) is Table 4 (decriptive statistics)? There equation (3) but equation (4) is missing...

Subsection 4.1.1 is very long and very specific to a normal reader; some tables are very difficult to read and understand.

Maybe you can add to the reference a publication about trade under risk (Broll, Mukherjee, Sensarma; Risk preference estimation of exporting firms; Scottish Journal of Political Economy, Vol. 67 (2010)).

Author Response

(The authors gave the same response as above.)

Round 2

Reviewer 2 Report

OK I agree with the other reviewers and have no further reservations.

Reviewer 3 Report

Dear Authors, 

The paper have been review but English language check is required. 

Best, 

Reviewer